# ELK3 Controls Gastric Cancer Cell Migration and Invasion by Regulating ECM Remodeling-Related Genes

**DOI:** 10.3390/ijms23073709

**Published:** 2022-03-28

**Authors:** Minwook Lee, Hyeon-Ju Cho, Kyung-Soon Park, Hae-Yun Jung

**Affiliations:** Department of Biomedical Science, College of Life Science, CHA University, Pangyo-Ro 335, Bundang-gu, Seongnam-si 463-400, Korea; minote95@naver.com (M.L.); whguswn94@naver.com (H.-J.C.)

**Keywords:** ETS transcription factor *ELK3*, gastric cancer metastasis, extracellular matrix remodeling, cell migration, cell invasion

## Abstract

Current therapeutic strategies for gastric cancer, including surgery and chemotherapy improve patient survival; however, the survival rate of patients with metastatic gastric cancer is very low. The molecular mechanisms underlying the dissemination of gastric cancer cells to distant organs are currently unknown. Here, we demonstrate that the E26 transformation-specific (ETS) transcription factor *ELK3* (*ELK3*) gene is required for the migration and invasion of gastric cancer cells. The *ELK3* gene modulates the expression of extracellular matrix (ECM) remodeling-related genes, such as bone morphogenetic protein (*BMP1*), lysyl oxidase like 2 (*LOXL2*), Snail family transcriptional repressor 1 (*SNAI1*), serpin family F member 1 (*SERPINF1*), decorin (*DCN*), and nidogen 1 (*NID1*) to facilitate cancer cell dissemination. Our in silico analyses indicated that *ELK3* expression was positively associated with these ECM remodeling-related genes in gastric cancer cells and patient samples. The high expressions of *ELK3* and other ECM remodeling-related genes were also closely associated with a worse prognosis of patients with gastric cancer. Collectively, these findings suggest that ELK3 acts as an important regulator of gastric cancer cell dissemination by regulating ECM remodeling.

## 1. Introduction

Gastric cancer is one of the most common types of cancer worldwide and a leading cause of cancer-related mortality [1,2]. The five-year survival rate of patients with gastric cancer with early-stage disease is 70%, whereas that of patients with invasive and distant metastasis is less than one third [3]. It is well-known that metastasis is closely associated with patient mortality. Several reports suggest that microRNAs, Wnt signaling, and extracellular matrix (ECM)-related factors are possible mechanisms of gastric cancer metastasis [4,5,6,7,8,9,10,11]. However, the molecular mechanisms underlying gastric cancer invasion and metastasis remain to be elucidated.

ETS transcription factor *ELK3* (*ELK3*) plays a pivotal role in promoting the progression and metastasis of a number of types of cancer, including breast, prostate, bladder and liver cancer [12,13,14,15,16]. *ELK3* is an E26 transformation-specific (ETS) transcription factor family member that regulates various genes, including zinc finger E-box binding homeobox 1 (*Zeb1*) and hypoxia-inducible factor-1α (*HIF1α*), to control cell migration and metastasis [14,15]. ELK3 also modulates cell migration by regulating the expression of extracellular matrix (ECM)-related genes, such as bone morphogenetic protein (*BMP1*), serpin family E member 1 (*SERPINE1*), and membrane type 1-matrix metalloproteinase (*MT1-MMP*) [17,18,19,20]. Of note, ECM remodeling-related proteins, including BMP1 and serpin family members, are involved in gastric cancer invasion and metastasis [7,21,22]. However, the functional link between the *ELK3* gene and ECM on gastric cancer cell migration is not yet fully understood, although the high expression of *ELK3* in patients with gastric cancer is closely associated with cancer progression [23].

Here, we investigated whether ELK3 expression is functionally associated with the metastatic phenotype of gastric cancer. We found that ELK3 regulated the migration and invasion of gastric cancer cells. RNA sequencing data revealed that ELK3 regulated other ECM remodeling-related genes, particularly bone morphogenetic protein (*BMP1*), lysyl oxidase like 2 (*LOXL2*), snail family transcriptional repressor 1 (*SNAI1*), serpin family F member 1 (*SERPINF1*), decorin (*DCN*), and nidogen 1 (*NID1*) for controlling cell dissemination. The analysis of various gastric cancer cell lines and gastric cancer patient samples also revealed a positive association between *ELK3* and *BMP1*, *LOXL2*, *SNAI1*, *SERPINF1*, *DCN*, and *NID1*, and these genes were associated with a poor prognosis. Our finding demonstrated that the *ELK3*-mediated genetic network in ECM remodeling contributed to the migration and invasion of gastric cancer cells.

## 2. Results

### 2.1. ELK3 Controls Gastric Cancer Cell Migration and Invasion in E-Cadherin Independent Manner

To examine the effect of ETS transcription factor ELK3 (ELK3) on the cell dissemination capacity of gastric cancer cells, we examined *ELK3* mRNA expression in various gastric cancer cell lines using the Cancer Dependency Map (DepMap) database (Figure 1A). We also analyzed *ELK3* mRNA level and protein level with five gastric cancer cell lines (Figure 1B,C). From these analyses and a previous report [24], we selected gastric cancer cell lines exhibiting a relatively low and high expression of the *ELK3* gene (SNU484 and SNU638 cells, respectively) and further examined the molecular link between ELK3 and cell migration.

First, we determined whether the overexpression of ELK3 regulated cell migration and invasion. The SNU484 cells exhibited overexpression of *ELK3* mRNA and protein after transfection with pLenti-cMyc-DDK-*ELK3* plasmid DNA (Figure 2A,B). The SNU484 cells overexpressing the *ELK3* gene exhibited a marked increase in cell migration and invasion (Figure 2C). Our previous study demonstrated that ELK3 repressed the expression of E-cadherin, a marker of epithelial cell, to trigger breast cancer metastasis [14]. Therefore, we examined the expression of epithelial-mesenchymal transition (EMT) markers in *ELK3* overexpressed SNU484 cells. Unlike the data from the previous study on breast cancer, E-cadherin expression was no different between control and ELK3 overexpressed SNU484 cells (Figure 2D). However, the Fibronectin protein levels were significantly increased by ELK3 in SNU484 cells. To confirm our finding, we transfected SNU638 cells with si*ELK3* to deplete the *ELK3* gene (Figure 3A,B). The depletion of *ELK3* significantly inhibited cell migration and invasion (Figure 3C). However, E-cadherin and Fibronectin expression were no different between control and *ELK3*-depleted SNU638 cells (Figure 3D), suggesting ELK3-mediated cell migration and invasion in gastric cancer cells is unnecessary for E-cadherin.

### 2.2. ELK3 Regulates Extracellular Matrix (ECM) Remodeling-Related Genes to Control Cell Migration and Invasion

To further investigate the molecular mechanisms underlying ELK3-mediated cell migration and invasion, RNA-Seq analysis of the control and *ELK3*-depleted SNU638 cells was performed. Among 25,000 genes, 262 genes were selected exhibiting a >1.5-fold change in expression, and a *p*-value < 0.05 was selected (Appendix A). Notably, the GSEA (Gene Set Enrichment Analysis) plot revealed that the *ELK3* depletion downregulated the expression of genes associated with cell junction disassembly and upregulated the genes involved in cell adhesion (Figure 4A). Figure 4B,C were analyzed by DAVID analysis using the 262 selected genes. The pie graph showed the categories of the top five of the biological pathway in the *ELK3*-depleted SNU638 cells compared with the control SNU638 cells (Figure 4B). Among them, we focused on the genes related to cell migration and adhesion. Heatmap showed 39 genes belonging to the category of cell migration and adhesion (Figure 4C). In particular, we were interested in 26 downregulated genes and these genes were very closely associated with cell migration and locomotion (Figure 4D). To further analyze the molecular network, we used STRING-based analysis using the 262 selected genes. BMP1-mediated ECM remodeling was one of the main molecular networks by ELK3-induced cell migration and adhesion (Figure 4E). Our bioinformatics analyses indicated that ELK3 regulated cell migration by modulating ECM remodeling-associated genes. After two different analyses, we selected six genes such as bone morphogenetic protein (*BMP1)*, lysyl oxidase like 2 (*LOXL2)*, Snail family transcriptional repressor 1 (*SNAI1)*, serpin family F member 1 (*SERPINF1)*, decorin (*DCN)*, and nidogen 1 (*NID1*) for further studies and performed RT-qPCR for validation of *ELK3*-related cell migration via those genes. The *ELK3*-depleted SNU638 cells exhibited a significant downregulation in the mRNA expression of *BMP1*, *LOXL2*, *SNAI1*, *SERPINF1*, *DCN*, and *NID1*. The overexpression of *ELK3* in *ELK3*-depleted SNU638 cells markedly increased the expression of these genes (Figure 5A), indicating that the *ELK3* gene positively regulates the expression of *BMP1*-mediated ECM remodeling-related genes in SNU638 cells. Finally, we determined whether ELK3 is essential for gastric cancer cell migration. Transwell assays revealed the inhibition of the migration of *ELK3*-depleted SNU638 cells; however, the cell migratory ability recovered after the *ELK3*-depleted SNU638 cells were transfected with the *ELK3* overexpression (Figure 5B). We performed an adhesion assay using collagen type I pre-coated plates to test the ability of binding of cancer cells to extracellular matrix (ECM) component after cell seeding. Notably, adhesion assay also revealed the markedly reduced adhesive ability of the *ELK3*-depleted SNU638 cells compared with the controls and *ELK3*-overexpressing cells (Figure 5C). To further study the molecular link between *ELK3* and *BMP1*-mediated ECM remodeling-related genes, we analyzed *ELK3* mRNA level after transfection with si*BMP1* in *ELK3*-depleted SNU638 with *ELK3* overexpression. In Figure 6A, *BMP1* depletion was insufficient for affecting the *ELK3* mRNA level although *BMP1* level was depleted, suggesting *ELK3* was an upstream regulator of the *BMP1* gene. Indeed, *BMP1* depletion significantly suppressed *LOXL2*, *SNAI1*, *SERPINF1*, *DCN*, and *NID1* genes after transfection with si*BMP1* in *ELK3*-depleted SNU638 with *ELK3* overexpression (Figure 6B). In addition, adhesion assay indicated significantly reduced cell adhesive ability in *BMP1* depleted SNU638 cells compared with *ELK3*-depleted SNU638 with *ELK3* overexpression (Figure 6C). Transwell and invasion assays also showed that *BMP1* depletion significantly decreased cell migration and invasion ability in *ELK3*-depleted SNU638 cells with *ELK3* overexpression (Figure 6D). These results clearly indicate that ELK3 is a key regulator of the migration and adhesion of gastric cancer cells and that it functions by regulating the expression of *BMP1*-mediated ECM remodeling-related genes.

### 2.3. ELK3 Positively Correlates with ECM Remodeling-Related Genes in Gastric Cancer Cells and Patient Samples

To confirm the positive correlation between *ELK3* and ECM remodeling-related genes in various gastric cancer cells and patient samples, in silico analyses were performed using databases from DepMap and TCGA. First, the correlation between *ELK3* expression and *BMP1*, *LOXL2*, *SNAI1*, *SERPINF1*, *DCN*, and *NID1* expression was analyzed in 19 different gastric cancer cell lines. The dot plots revealed a significant positive correlation between the expression of the *ELK3* and *LOXL2* genes; however, this positive correlation was not significant for the other genes (Figure 7A). Notably, the analysis of patients with gastric cancer indicated the marked positive regulation of *ELK3* and *BMP1*, *LOXL2*, *SNAI1*, *SERPINF1*, *DCN*, and *NID1* gene expression (Figure 7B), suggesting a significant positive correlation between *ELK3* and ECM remodeling-associated genes in gastric cancer. To further examine the role of these genes in gastric cancer progression, their expression levels were compared between healthy tissue and gastric cancer tissue. The analysis of data from TCGA and GTEx revealed that *ELK3* and *BMP1*, and *LOXL2* were highly expressed in the gastric cancer samples compared with the healthy samples (Figure 7C). The *SNAI1* gene also exhibited a higher expression in the gastric cancer samples, although this increase was not statistically significant. No differences were observed in *SERPINF1*, *DCN*, and *NID1* expression between the healthy samples and cancer patient samples. Subsequently, the clinical relevance between the *ELK3* gene and ECM remodeling-related genes and the prognosis of patients with gastric cancer were examined using Kaplan-Meier survival plots. The overall survival of patients with gastric cancer with a high expression of *ELK3* was significantly lower than that of patients with a low expression. A similar pattern was observed in patients with a high expression of ECM remodeling-related genes. Patients with gastric cancer with a high expression of *BMP1*, *LOXL2*, *SNAI1*, *SERPINF1*, *DCN*, and *NID1* had a poor prognosis (Figure 7D). The data from in silico analyses indicate that *ELK3* positively regulate the expression of ECM remodeling-related genes, resulting in a poor prognosis of patients with gastric cancer. Thus, this gene signature might be a surrogate marker for gastric cancer progression.

## 3. Discussion

*ELK3* is highly expressed in patients with gastric cancer; however, the association between *ELK3* expression and gastric cancer progression remains unknown. Our findings demonstrated that *ELK3* promoted gastric cancer cell migration and invasion. *ELK3* also regulated the expression of extracellular matrix (ECM) remodeling-related genes, thereby enhancing cancer cell dissemination. The gene analysis of samples from patients with gastric cancer indicated that a high *ELK3* expression is positively correlated with *BMP1*, *LOXL2*, *SNAI1*, *SERPINF1*, *DCN*, and *NID1* expression, all of which are closely associated with a poor prognosis.

ELK3 is a transcription factor that can cooperate with numerous partners to control cancer metastasis. Previous studies have demonstrated that ELK3 regulates breast cancer metastasis through *Zeb-1*, *MT1-MMP*, and *GATA3* gene-mediated signaling pathways [14,25,26]. Another study reported that a blocking agent suppressed Ras/Erk-ELK3 signaling and also inhibited the progression of prostate cancer [16], suggesting that ELK3 is a promising target molecule for preventing cancer metastasis. However, evidence of ELK3-mediated metastasis in numerous cancer types is lacking. Our study identified a molecular network of *ELK3*-mediated gastric cancer cell migration and invasion, the underlying mechanism of which is the regulation of ECM remodeling.

The ECM, which comprises numerous proteins, including collagens, glycoproteins, and secreted proteins, interacts with cells and delivers extracellular signals that can alter the cellular phenotype [27,28,29]. ECM remodeling leads to a change in the microenvironment, which contributes to cancer progression and metastasis [30]. The LOX family of oxidases controls ECM structural components and accelerates cancer metastasis by altering the tumor microenvironment [31,32]. LOXL2 also promotes EMT-mediated metastasis by stabilizing the SNAI1 protein [33]. NID1 is a basement membrane glycoprotein that expedites breast lung metastasis [34]. Several groups suggest that genes associated with ECM remodeling are involved in promoting metastasis; however, the detailed molecular mechanisms that regulate gene expression remain unclear. The present study demonstrated that ELK3 regulated the expression of certain ECM remodeling-related genes to induce gastric cancer cell migration and invasion.

In conclusion, our finding suggests that ELK3 contributes to gastric cancer cell dissemination by regulating the expression of ECM remodeling-related genes. The *ELK3* expression in gastric cancer cells and human gastric cancer samples positively correlated with that of *BMP1*, *LOXL2*, *SNAI1*, *SERPINF1*, *DCN*, and *NID1*, suggesting that this gene signature may be predictive molecular markers for aggressive gastric cancer progression.

## 4. Materials and Methods

### 4.1. Cell Lines and Cell Culture

The human gastric cancer cell lines, MKN1, SNU216, SNU484, SNU638, and SNU668 were obtained from the Korean Cell Line Bank (Seoul, South Korea). All gastric cancer cell lines were cultured in RPMI-1640 medium containing 25 mM HEPES, 25 mM NaHCO3, and 300 mg/L L-glutamine (Thermo Fisher Scientific, Inc., Waltham, MA, USA) with 10% FBS (Thermo Fisher Scientific, Inc.) and 1% penicillin/streptomycin (Thermo Fisher Scientific, Inc.). 293T cell was obtained from American Type Culture Collection (ATCC, Manassas, VA, USA). 293T cell was cultured in DMEM medium (Thermo Fisher Scientific, Inc.) with 10% FBS and 1% penicillin/streptomycin. All cell lines were cultured at 37 °C with 5% CO_2_.

### 4.2. Plasmid DNA, Small Interfering (si)RNA Transfection, and Short Haripin (sh)RNA Transduction

SNU638 cells (2 × 10^5^ cells/mL) were seeded in a 60-mm dish and transfected with 50 nM si*ELK3* or siCon using Lipofectamine 2000^®^ (Invitrogen; Thermo Fisher Scientific, Inc.). SMART Pool siRNA targeting *ELK3* (#L-010320-00-0005) was purchased from Dharmacon, Inc. (Lafayette, CO, USA), SMART Pool siRNA targeting *BMP1* (#O-220221-0056) and non-specific siRNA (sn-1003) were purchased from Bioneer Corporation (Daejeon, South Korea). For overexpress *ELK3* gene, SNU484 cells (5 × 10^5^) were seeded in a 60-mm dish and transfected with 2.5 µg of con and 2.5 µg of pLenti-cMyc-DDK-*ELK3* using Lipofectamine 2000^®^ reagent. pLenti-cMyc-DDK-*ELK3* was generated from cDNA by PCR. The protein coding region of *ELK3* gene was ligated into the pLenti-cMyc-DDK vector (PS1000064; OriGene Technologies, Inc., Rockville, MD, USA). The transfected cells were incubated in 37 °C for 48 h to 72 h. The efficacy of transient transfection was approximately 30% in SNU638 cells. To generate sh*ELK3* lenti-virus, we used a shRNA targeting 3′UTR of *ELK3* (RHS4531-EG2004) from Dharmacon, Inc. The sequence of sh*ELK3* against the *ELK3* gene was 5′-TTTCATCAGTTAATGAGTC-3′. 2 × 10^6^ of 293T cells were seeded in a 100-mm culture dish and transfected with 5.0 µg of sh*ELK3*, 4.6 µg of pCMV delta8.2 and 0.5 µg of VSVG for 48 h at 37 °C/5% CO_2_ incubator. After incubation, harvested lenti-virus was applied to target cells. After 2.5 µg/mL of puromycin selection, our lenti-virus titer was around 50% by FACS analysis.

### 4.3. Reverse Transcription-Quantitative PCR (RT-qPCR)

Total RNA was extracted from the cells using TRIzol reagent (Invitrogen; Thermo Fisher Scientific, Inc.). Reverse transcription was conducted using SuperScript™ II Reverse Transcriptase (Invitrogen; Thermo Fisher Scientific, Inc.) and temperature protocol was as follows; 25 °C for 10 min, 42 °C for 60 min, 72 °C for 10 min. qPCR was performed using TOPreal™ qPCR 2X PreMIX (Enzynomics, Inc., Daejeon, South Korea) and temperature protocol is as follows; 95 °C for 10 min and 15 s, 60 °C for 30 s, 72 °C for 30 s. The mRNA expression levels were analyzed using the 2-ΔΔCq [35] method and normalized against *GAPDH*. The sequences of the primers are listed in Table 1.

### 4.4. Cell Migration and Invasion Assays

Cell migration and invasion were analyzed using a 24-well Transwell insert (a poly carbonate membrane with a pore size of 8.0 μm; Corning, Inc., Corning, NY, USA). For the invasion assay, the upper surface of the membrane was coated with Matrigel (BD Biosciences, San Jose, CA, USA) and 2 × 10^4^ SNU638 cells were seeded into the top chamber, which was then filled with 100 μL serum-free RPMI-1640 medium. Subsequently, 700 μL complete medium were placed into the bottom chamber, and the Transwell assay kit was incubated for 24 h at 37 °C. Migrated or invaded cells on the lower surface of the insert filter were fixed with 4% paraformaldehyde and stained with crystal violet (Sigma-Aldrich; Merck KGaA, Burlington, MA, USA) at room temperature for 30 min. The cells were photographed under an optical microscope (Nikon Corporation, Tokyo, Japan).

### 4.5. Cell Adhesion Assay

A 96-well plate was firstly coated with type I collagen for 1 h at 37 °C. Cells (4 × 10^5^) were seeded on the plate and incubated at 37 °C in a 5% CO_2_ incubator for 30 min. After washing out the non-attached cells, the remaining cells were stained with crystal violet (Sigma-Aldrich; Merck KGaA) for 10 min at room temperature and analyzed under an optical microscope (Nikon Corporation). For the quantification analysis, 2% SDS was added to the remaining cells and measured in 550 nm using a microplate reader.

### 4.6. Western Blot Analysis

The cells were lysed with cell lysis buffer (Cell Signaling Technology, Inc., Danvers, MA, USA). A total of 40 µg of proteins are separated by sodium dodecyl sulfate-polyacrylamide gel electrophoresis (SDS-PAGE) and transferred to polyvinylidene difluoride membranes (Bio-Rad Laboratories, Inc., Hercules, CA, USA). The membranes were then blocked with 4% of BSA (Morebio, Inc., Hanam, South Korea) in TBS-T (0.1% Tween) in RT for 1 h for ELK3 protein and 2% BSA in TBS-T (0.1% Tween) in RT for 1 h for GAPDH, Fibronectin and E-cadherin protein. The membranes were probed overnight at 4 °C with anti-ELK3 (NBP2-01264; 1:1000, Novus Biologicals, LLC, Minneapolis, MN, USA), anti-E-cadherin (sc-7870; 1:1000, Santa Cruz Biotechnology, Inc. Dallas, TX, USA), anti-Fibronectin (F3648; 1:1000, Sigma-Aldrich) or anti-GAPDH (sc-166574; 1:1000, Santa Cruz Biotechnology, Inc.) antibodies, and then washed with TBS-T (0.1% Tween). After washing, the membranes were incubated for 1 h at room temperature with an anti-mouse secondary antibody (GTX213111-01; 1:4000, GeneTex, Inc., Irvine, CA, USA), or an anti-rabbit secondary antibody (GTX213110-01; 1:4000, GeneTex, Inc.). Immunoreactivity was detected using an ECL kit (Thermo Fisher Scientific, Inc.) and ImageQuant Las 4000 (GE Healthcare, Chicago, IL, USA).

### 4.7. RNA Sequencing (RNA-Seq) Analysis

RNA-Seq analysis was performed by Ebiogen, Inc. (Seoul, South Korea). Briefly, a total library was constructed using a QuantSeq 3′ mRNA-Seq Library Prep kit (Lexogen, Wien, Austria). Following PCR, high-throughput single-end 75 bp sequencing was performed using a NextSeq 500 apparatus (Illumina, Inc., San Diego, CA, USA). During GEO analysis, gene sets exhibiting a fold change in expression > 1.50 and *p*-values < 0.05 were considered statistically significant. Gene expression analysis and Gene Ontology analysis were performed using the Excel-based Differentially Expressed Gene Analysis (ExDEGA) software package and the Database for Annotation, Visualization, and Integrated Discovery (DAVID) tools, last accessed at 29 April 2021 (https://david.ncifcrf.gov/). Gene clustering analysis was performed using MeV (version 4.9.0). Gene Set Enrichment Analysis (GSEA) was performed using GSEA software, last accessed at 8 June 2021 (version 4.1.0) (https://www.gsea-msigdb.org/gsea/index.jsp). Protein–protein interaction was evaluated using STRING, last accessed at 25 February 2022 (Search Tool for the Retrieval of Interacting Genes) tool [36]. Gene set collections were downloaded from the Molecular Signatures Database download page. The data discussed in the present study have been deposited in NCBI Gene Expression Omnibus [37] and are accessible through GEO Series accession No. GSE181840, last accessed at 14 September 2021 (https://www.ncbi.nlm.nih.gov/geo/query/acc.cgi?acc=GSE181840).

### 4.8. Genomic Analyses of Gastric Cancer Cell Lines and Samples from Patients with Gastric Cancer

Gene expression analysis of 408 samples from patients with gastric cancer and 211 healthy samples from the TCGA and GTEx databases were performed on the Gene Expression Profiling Interactive Analysis 2 (GEPIA2) website, last accessed at 24 February 2022 (http://gepia2.cancer-pku.cn). Spearman’s correlation coefficient was calculated, and a *p*-value < 0.05 was considered to indicate a statistically significant difference. Survival curves were analyzed using a Kaplan–Meier plotter. Gene expression analysis data for the gastric cancer cell lines were downloaded from the Cancer Dependency Map, last accessed at 20 February 2022 (DepMap). The correlation analyses were based on data from 19 gastric cancer cell lines, excluding high or low normalized expression values for each gene.

### 4.9. Statistical Analysis

All results are derived from at least three independent experiments. Statistical analysis was performed using GraphPad Prism 7.0 software. *p*-values were calculated using a student’s unpaired *t*-test or one-way ANOVA. *p* < 0.05 was considered to indicate a statistically significant difference. Error bars indicate the standard deviation.

## Figures and Tables

**Figure 1 ijms-23-03709-f001:**
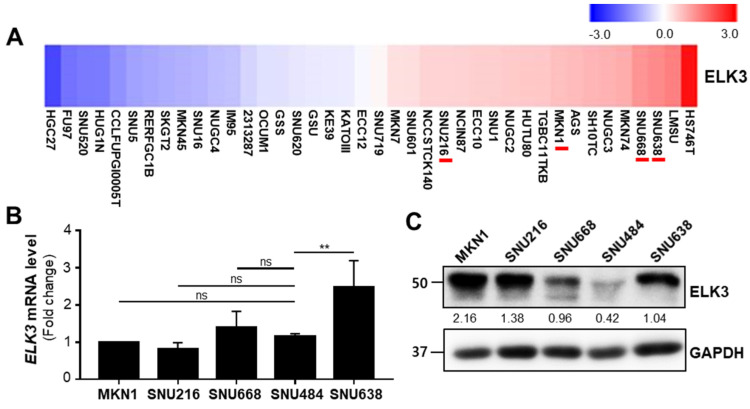
ETS transcription factor ELK3 (ELK3) expression pattern in various gastric cancer cell lines. (**A**) *ELK3* expression in gastric cancer cell lines from the Cancer Dependency Map (DepMap) database. (**B**) *ELK3* mRNA expression in five gastric cancer cell lines, measured by reverse transcription-quantitative PCR. Error bars represent the standard deviation; ** *p* < 0.01 (one-way ANOVA). (**C**) ELK3 protein levels in five gastric cancer cell lines. The values indicate the signal intensities of ELK3 relative to GAPDH.

**Figure 2 ijms-23-03709-f002:**
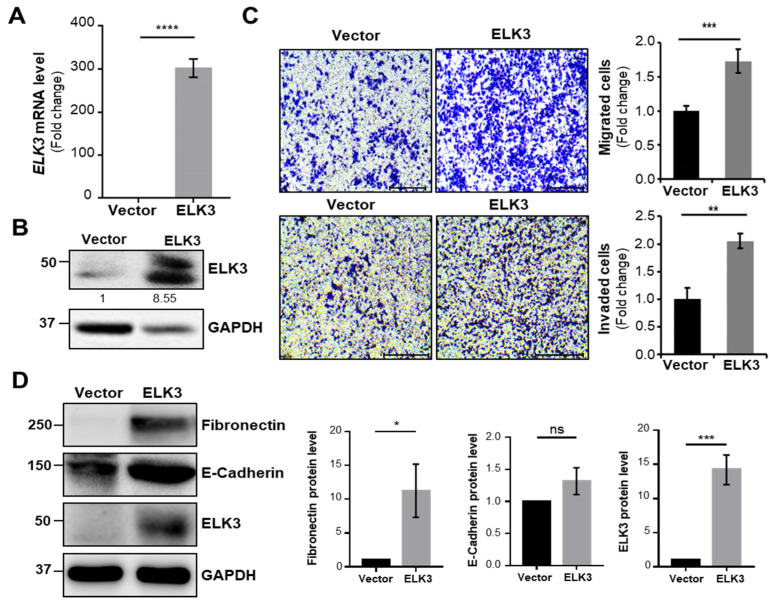
ELK3 increases the migration and invasion of SNU484 cell line in E-cadherin independent manner. (**A**) *ELK3* mRNA levels in SNU484 cells transfected with control vector and pLenti-cMyc-DDK-*ELK3* plasmid. Error bars represent the standard deviation; **** *p* < 0.0001 (student’s *t*-test). (**B**) ELK3 protein levels in SNU484 cells transfected with control vector and pLenti-cMyc-DDK-*ELK3* plasmid. The values indicate the signal intensities of ELK3 relative to GAPDH. (**C**) Representative images showing cell migration (top two panels) and cell invasion (bottom two panels) of SNU484 cells transfected with control vector and pLenti-cMyc-DDK-*ELK3* plasmid. Scale bar, 200 μm. Bar graphs indicate the number of migrated cells and invaded cells. Error bars represent the standard deviation; ** *p* < 0.01 and *** *p* < 0.001 (student’s *t*-test). (**D**) Epithelial marker (E-cadherin), mesenchymal marker (Fibronectin), and ELK3 protein expression in SNU484 cells transfected with control vector and *ELK3*. Bar graphs indicate the quantified levels of each protein, in relative scales. Error bars represent the standard deviation; * *p* < 0.05 and *** *p* < 0.001 (student’s *t*-test). ELK3, ETS transcription factor ELK3; Vector, control vector.

**Figure 3 ijms-23-03709-f003:**
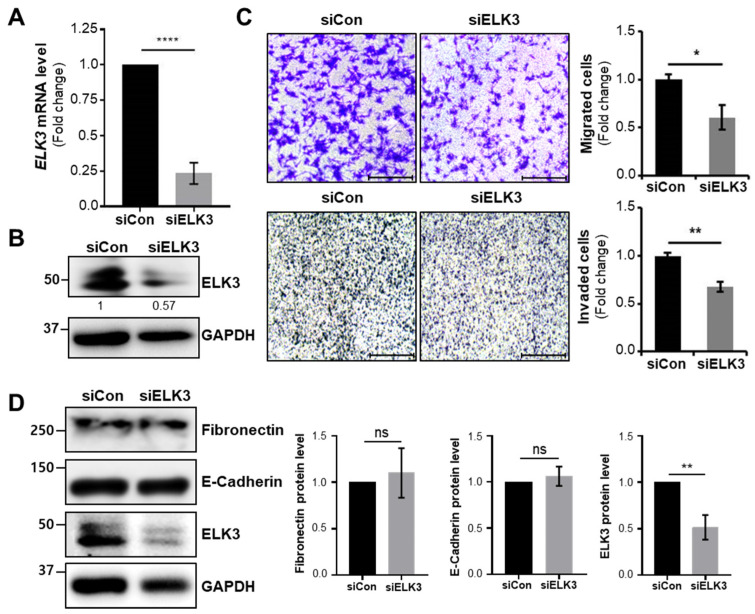
Depletion of ELK3 decreases the migration and invasion of SNU638 cell line in E-cadherin independent manner. (**A**) *ELK3* mRNA levels in SNU638 cells transfected with siCon or si*ELK3*. Error bars represent the standard deviation; **** *p* < 0.0001 (student’s *t*-test). (**B**) ELK3 protein levels in SNU638 cells transfected with siCon or si*ELK3*. The values indicate the signal intensities of ELK3 relative to GAPDH. (**C**) Representative images showing cell migration (top two panels) and cell invasion (bottom two panels) of SNU638 cells transfected with siCon or si*ELK3*. Scale bar, 200 μm. Bar graphs indicate the number of migrated and invaded cells, respectively. Error bars represent the standard deviation; * *p* < 0.05 and ** *p* < 0.01 (student’s *t*-test). (**D**) Epithelial marker (E-cadherin), mesenchymal marker (Fibronectin), and ELK3 protein expression in SNU638 cells transfected with siCon or si*ELK3*. Bar graphs indicate the quantified levels of each protein, in relative scales. Error bars represent the standard deviation; ** *p* < 0.01 (student’s *t*-test). ELK3, ETS transcription factor ELK3; si, small interfering RNA; Con, control.

**Figure 4 ijms-23-03709-f004:**
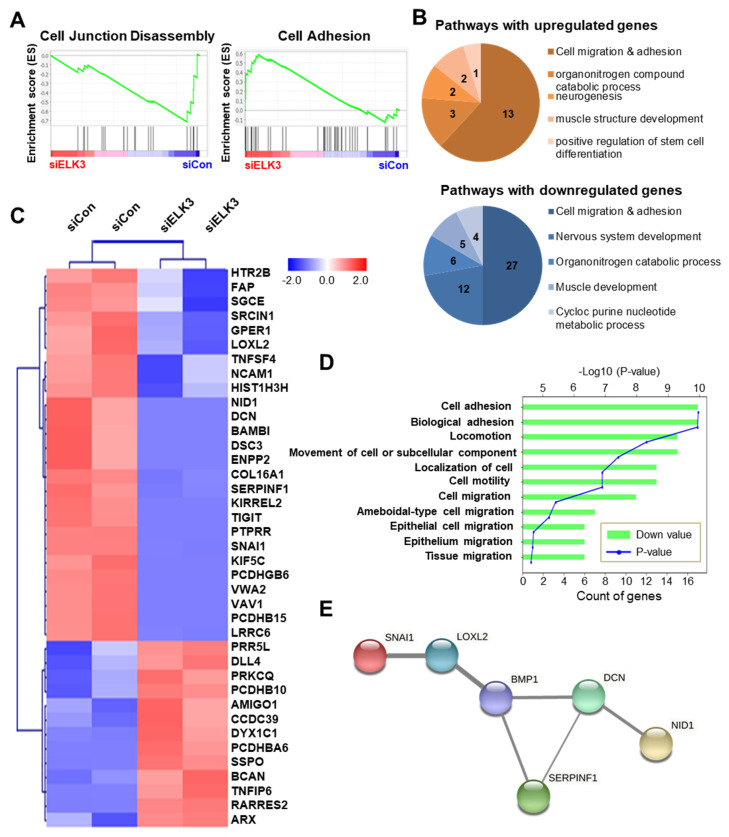
ELK3 controls extracellular matrix (ECM) remodeling-related genes to control cell migration. (**A**) Gene set enrichment analysis of RNA sequencing data from SNU638 cells transfected with siCon or si*ELK3*. Cell adhesion-related genes were positively enriched and cell junction disassembly-related genes were negatively enriched in SNU638 cells transfected with si*ELK3*. (**B**) Pie graphs showing biological pathways in *ELK3*-depleted SNU638 cells. Numbers indicate the number of genes in each biological pathway. (**C**) Heatmap analysis of genes associated with cell migration and cell adhesion. (**D**) Bar graph showing the Database for Annotation, Visualization, and Integrated Discovery (DAVID) annotation of genes involved in cell migration and cell adhesion. (**E**) Map data of Search Tool for the Retrieval of Interacting Genes analysis (STRING). The thickness of the lines represents the confidence of the interaction.

**Figure 5 ijms-23-03709-f005:**
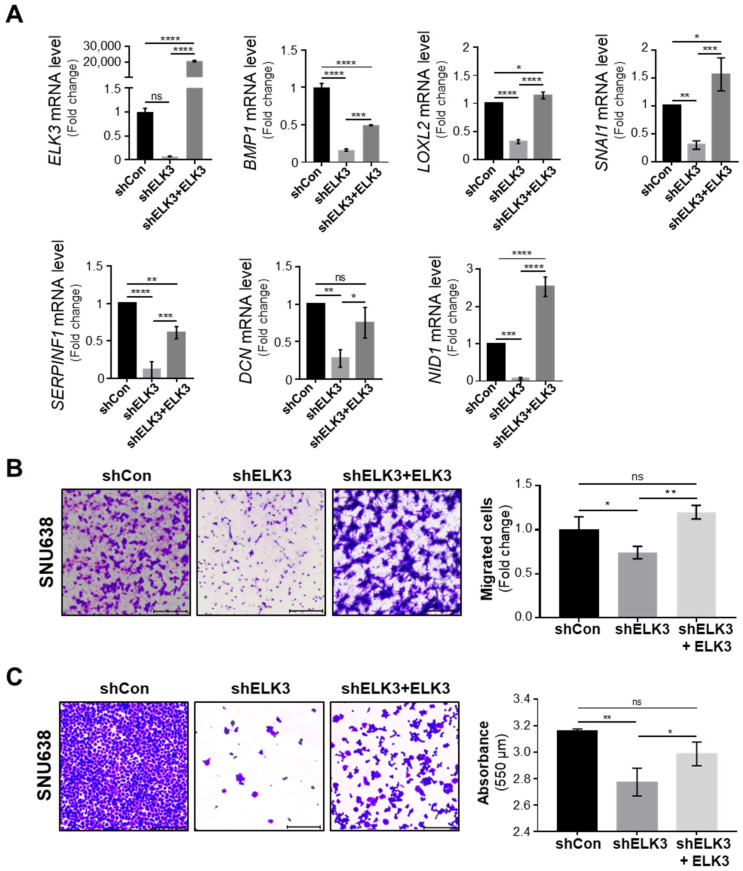
*ELK3* is required for gastric cancer cell migration and adhesion. (**A**) Expression of mRNA encoding the indicated genes in SNU638 cells transfected with shCon, sh*ELK3*, or sh*ELK3* plus pLenti-cMyc-DDK-*ELK3* plasmid DNA. Error bars represent the standard deviation; * *p* < 0.05, ** *p* < 0.01, *** *p* < 0.001, and **** *p* < 0.0001 (one-way ANOVA). (**B**) Representative images showing the cell migration of SNU638 cells transfected with shCon, sh*ELK3*, or sh*ELK3* plus pLenti-cMyc-DDK-*ELK3* plasmid DNA. Scale bar, 200 μm. Bar graphs indicate the number of migrated cells. Error bars represent the standard deviation; * *p* < 0.05 and ** *p* < 0.01 (one-way ANOVA). (**C**) Representative images showing cell adhesion of SNU638 cells transfected with shCon, sh*ELK3*, or sh*ELK3* plus pLenti-cMyc-DDK-*ELK3* plasmid DNA. Scale bar, 200 μm. Bar graphs indicate the absorbance value of adherent cells. Error bars represent the standard deviation; * *p* < 0.05 and ** *p* < 0.01 (one-way ANOVA). ELK3, ETS transcription factor ELK3; sh, short hairpin RNA; Con, control.

**Figure 6 ijms-23-03709-f006:**
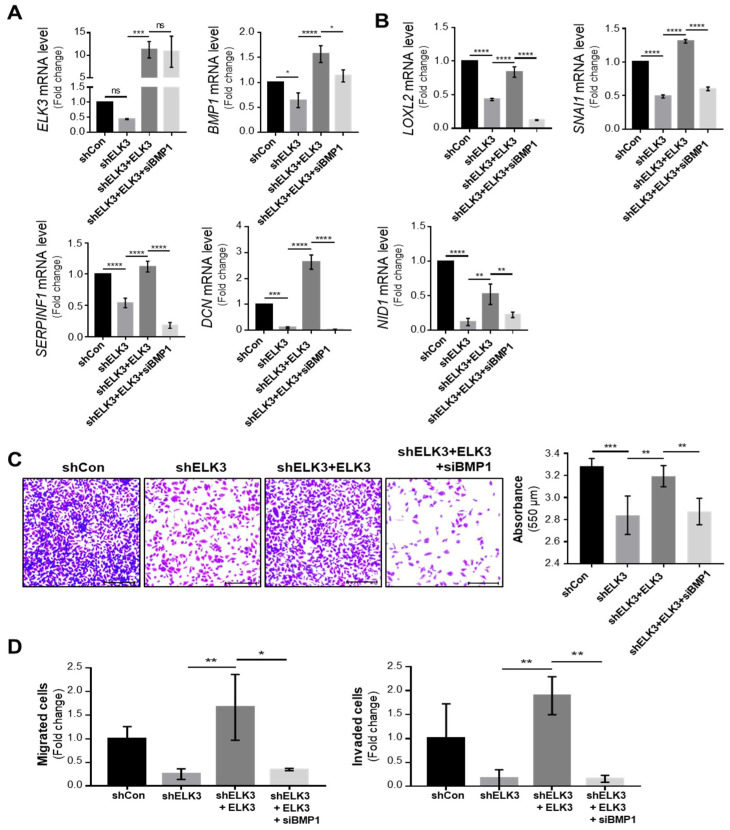
*ELK3* regulates gastric cancer cell adhesion through bone morphogenetic protein1 (BMP1)-mediated ECM remodeling. (**A**) Expression of mRNA encoding the *ELK3* and *BMP1* genes in SNU638 cells transfected with shCon, sh*ELK3*, sh*ELK3* plus pLenti-cMyc-DDK-*ELK3* plasmid DNA (*ELK3* rescue), or *ELK3* rescue plus si*BMP1*. Error bars represent the standard deviation; * *p* < 0.05, *** *p* < 0.001, and **** *p* < 0.0001 (one-way ANOVA). (**B**) Expression of mRNA encoding the *LOXL2, SNAI1, SERPINF1, DCN,* and *NID1* genes in SNU638 cells transfected with shCon, sh*ELK3*, *ELK3* rescue, or *ELK3* rescue plus si*BMP1*. Error bars represent the standard deviation; ** *p* < 0.01, *** *p* < 0.001, and **** *p* < 0.0001 (one-way ANOVA). (**C**) Representative images showing cell adhesion of SNU638 cells transfected with shCon, sh*ELK3*, *ELK3* rescue, or *ELK3* rescue plus si*BMP1*. Scale bar, 200 μm. Bar graphs indicate the absorbance value of adherent cells. Error bars represent the standard deviation; ** *p* < 0.01 and *** *p* < 0.001 (one-way ANOVA). (**D**) Representative images showing cell migration and invasion of SNU638 cells transfected shCon, sh*ELK3*, *ELK3* rescue, or *ELK3* rescue plus si*BMP1*. Scale bar, 200 μm. Bar graphs indicate the number of migrated cells. Error bars represent the standard deviation; * *p* < 0.05 and ** *p* < 0.01 (one-way ANOVA). ELK3, ETS transcription factor ELK3; sh, short hairpin RNA; Con, control.

**Figure 7 ijms-23-03709-f007:**
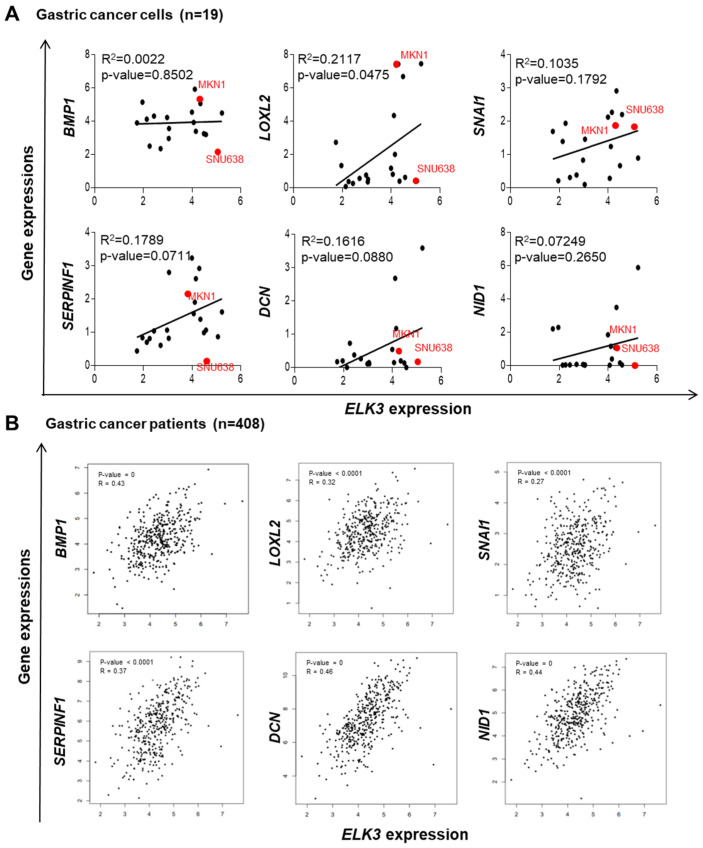
*ELK3* positively regulates ECM remodeling-related genes in gastric cancer cells and patient tissues. (**A**) Dot plots illustrating the correlation between *ELK3* and the indicated genes. Each dot denotes a single cell line. Red dots denote SNU638 and MKN1 cells used in the present study. All cell lines expression data were obtained from the Cancer Dependency Map database. *p* < 0.05 was considered to indicate a statistically significant difference. (**B**) Correlation plot illustrating the link between *ELK3* and the indicated genes in patients with gastric cancer. Each dot denotes a single patient. *p* < 0.05 was considered to indicate a statistically significant difference. (**C**) Box plots illustrating the expression of the indicated genes in samples from patients with gastric cancer (n = 408) and healthy samples (n = 211). *p* < 0.05 was considered to indicate a statistically significant difference. (**D**) Kaplan-Meier survival plots showing the overall survival of patients with gastric cancer with high or low expression of the indicated genes. All patient data were obtained from the Cancer Genome Atlas. *p* < 0.05 was considered to indicate a statistically significant difference. STAD, stomach adenocarcinoma.

**Table 1 ijms-23-03709-t001:** The primer sets for RT-qPCR.

Target	Sequence
*ELK3*	F: 5′-ACCCAAAGGCTTGGAAATCT-3′
	R: 5′-TGTATGCTGGAGAGCAGTGG-3′
*BMP1*	F: 5′-TGGCCGACTACACCTATGAC-3′
	R: 5′-GGAGGACTTACGAGCTGTGT-3′
*LOXL2*	F: 5′-GGACATGTACCGCCATGACA-3′
	R: 5′-ATAGCGGCTCCTGCATTTCA-3′
*SNAI1*	F: 5′-GCAAATACTGCAACAAGG-3′
	R: 5′-GCACTGGTACTTCTTGACA-3′
*SERPINF1*	F: 5′-GCTATGACCTGTACCGGGTG-3′
	R: 5′-GTCTGGGCTGCTGATCAAGT-3′
*DCN*	F: 5′-AATTGAAAATGGGGCTTTCC-3′
	R: 5′-GCCATTGTCAACAGCAGAGA-3′
*NID1*	F: 5′-CCTGCTACATCGGCACTCAT-3′
	R: 5′-TGAAAACTGGTAGCCCTCCAC-3′
*GAPDH*	F: 5′-GGGTGTGAACCATGAGAA-3′
	R: 5′-GTCTTCTGGGTGGCAGTGAT-3′

## Data Availability

The datasets used and/or analyzed during the current study are available from the corresponding author upon reasonable request. The data accessible through GEO Series accession number GSE181840 (https://www.ncbi.nlm.nih.gov/geo/query/acc.cgi?acc=GSE181840 (accessed on 14th. September 2021)).

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
