# Peer review of "ELK3 Controls Gastric Cancer Cell Migration and Invasion by Regulating ECM Remodeling-Related Genes"

_ijms, 2022, doi:10.3390/ijms23073709_

Round 1

Reviewer 1 Report

This manuscript by Lee and co-workers reports results from experiments designed to investigate possible mechanisms underlying the development of metastatic gastric cancer, which is known to affect about 40 % of gastric cancer patients. Through a combination of cell culture and in silico approaches, the authors describe that alterations in the expression of ELK3, a member of the ETS family of transcription factors, correlates with changes in the expression of genes encoding proteins functionally related to the remodeling of the extracellular matrix (ECM), and these changes are also reported found to be associated with a poor prognosis for gastric cancer patients. These data are taken as the basis to conclude that ELK3 is an important regulator of the dissemination of gastric cancer cells, presumably leading to the metastatic phenotype.

Taking into consideration that metastasis away from the primary site of tumors is what ultimately causes the death of most cancer patients, investigations on molecular mechanisms of metastasis are certainly worthwhile. In this regard, the main research goal of the present manuscript is clearly meritorious. However, there are a number of concerns (see below) that seriously decrease the enthusiasm towards the soundness of the reported findings and conclusions.

Major concerns:

(1) The involvement of ELK3 in gastric cancer has been reported already, including cell culture data showing that ELK3 downregulation inhibited gastric cancer cell proliferation and induced their apoptotic death, thereby demonstrating anti-tumor activity.

(2) Data in Figure 1 appear to be somewhat incomplete. SNU484 cells, which were selected by the authors as the model for low ELK3 expression, are not included in panel A. In addition, by comparing panels B and C, it becomes immediately obvious that there is no correlation between the levels of mRNA and protein expressed by the various cells lines, which creates doubts about data focusing on mRNA analyses later in the paper (particularly those related to the analysis of patient samples, as there is no way to know that corresponding levels of the proteins change in the same up or down direction. In panel C, the exposure presented for the GAPDH loading controls makes them appear as saturated in all lanes (a problem common to Figure 2 and 3), which prevents an accurate estimation of the expression differences of the proteins under investigation. Finally, the legend refers to data from six (6) cell lines in panel C, when there are only five (5) cell lines included in fact.

(3) With regard to Figure 2, the authors should indicate the reason why there are two ELK3 protein bands in panels A and D, which were not detected for the same cells lines in Figure 1C. It is surprising that what look like substantial changes in ELK3 protein expression (4.22-fold increase in panel B, and nearly 2.4-fold reduction in panel D) only lead to relatively modest changes (less than 2-fold in most cases) in the migration and invasion by the two cell lines. Because these experiments were presumably performed following transient transfections (up to 48 t0 72 h), it becomes important, particularly in experiments using siRNA, that the authors indicate the percentage of cells that were actually transfected, to see whether the apparently modest phenotypic changes observed were due to the presence of large number of untransfected cells in the cell populations used in the experiments. The saturation of the loading protein controls is also a problem in this Figure.

(4) The title of the legend for Figure 3 should be modified, because it does not really show any data on migration. Also, the lack of correlation between the levels of E-cadherin in the blot and those in the corresponding histogram (panel A), as well as those between the level of ELK3 protein in vector transfected SNU484 cells (panel A) and those shown in Fig. 1C, plus the changes for the siRNA-mediated downregulation on the levels of ELK3 protein observed in panel B and those shown in Fig. 2E, all together suggest that most likely the data derived from the use of more than one batch of transfected cells, as a source of inconsistencies and variability. Surprisingly, the authors do not seem to vive any importance to the dramatic increase in the expression of fibronectin (a very relevant ECM component) in SNU484 cells overexpressing ELK3 (panel A). Finally, as indicated above, the saturation of the loading protein controls is also a problem in this Figure.

(5) In relation to Figure 4, the text talks about seven (7) genes that were selected from the in silico analyses, but it only lists six (6). However, BMP1 is included in this list of six genes, despite the fact that it does not even appear in panel C. The authors need to indicate the basis for the selection of those 6 genes out of the 27 indicated in panels B and C over any other possible functional links that could be established. Also, it is not clear why the authors focus on downregulated genes and do not seem to care about the upregulated genes in cells overexpressing ELK3, and that is particularly striking in the context of the dramatic increase in fibronectin described in Figure 3.

(6) In order to properly interpret the data in Figure 5, the authors must include two important pieces of information: (a) the target sequence for the shELK3 used and its location in the ELK3 mRNA, and (b) how the ELK3 sequence in the lentiviral vector was modified so that it would not include the shELK3 target sequence to prevent its downregulation in ELK3 reconstitution experiments. In addition, in panel A (and in other locations in the manuscript) the authors refer to Serpin E1 instead of F1, and that must be taken care of everywhere. Finally, the simultaneous inhibition of cell adhesion and migration seems to be rather contradictory, as one would expect that migration should be easier for cells with reduced adhesion capacity, yet the authors do not even comment or discuss about this issue.

(7) Figure 7 (panels A, B and C) presumably deal with differences in mRNA expression, although the legend fails to adequately clarify that point. But it was clear from Figure 1 that for ELK3 there was no correlation between the levels of mRNA and those of protein, which are the one truly providing functional information. On that basis, the data provided in panels A, B and C become rather uninterpretable. The authors need to provide information on the criteria/method used to establish the cut-off point between “high” and “low” expression in panel D.

(8) In the last paragraph of the “Discussion” the authors state that their findings suggest that “ELK3 triggers gastric cancer cell dissemination”. The use of the term “triggers” is inappropriate because it has an “initiating” connotation, and the data presented in the manuscript do not even address whether ELK3 is involved as an initiator or simply as a contributor to the maintenance of a more invasive phenotype. Therefore, the statement should be modified.

(9) The authors should follow the internationally accepted nomenclature rules that indicate that the symbols for nucleic acids (genes, DNAs, mRNAs or other RNAs) should be identified using the italics font, whereas the same symbols should be presented using a regular font when used to refer to their protein products. In several parts of the manuscript it is not clear whether the authors are talking about the genes of the proteins.  The authors should also distinguish between genes and proteins when referring to functional roles: for instance, it is the ELK3 protein not the gene (as it is stated in the Abstract) that modulates the expression of ECM-remodeling elated genes. This is an essential conceptual notion that should not be misused.

In addition to the points indicated above, the manuscript contains a number of typographical errors that should be corrected: “than”, instead of “then”; “2x10Y cells” instead of “2x10Y cells” (several times; “oC” instead of “⁰C”; and others.

Reviewer 2 Report

Lee M et al. have characterized the functional role of ELK3 transcription factor in gastric cancer cells in the manuscript entitled “ELK3 controls gastric cancer cell migration and invasion by regulating ECM remodeling-related genes”. Their data suggest that ELK3 regulates invasion and migration. Mechanistically, RNA seq analysis of knockdown cells shows that ELK3 regulates ECM remodeling-related genes. The association is found also in silico with human gastric cancer samples.  In addition, in silico data shows that ELK3 as well as some of the ECM remodeling-related genes that it regulates, associate with poor prognosis in gastric cancer patients. The topic is interesting. Comments:

ELK3 is suggested to regulate invasion and cell migration via regulation of select ECM remodeling-related genes. There is no experimental data supporting these links. The phenotypes are however observed in experimental models without intact microenvironment and without real ECM. The authors should explain this discrepancy and provide some evidence to support the claims.

Fig 3. More different EMT markers should be looked in addition to E-cadherin and Fibronectin. Was there any change in EMT markers in the RNA seq data (Fig 4)? The use of some factor or condition stimulating EMT could be used in the presence or absence of ELK3 to uncover its role in EMT.

Fig 4B shows that genes regulating cell migration and adhesion were both upregulate and downregulated. This is confusing.

More unbiased ELK3-regulated networks related to invasion, EMT and metastasis (not only preselected BMP1, LOXL2, SNAI1, SERPINF1, DCN, and NID1) should be analyzed in human samples in silico with bioinformatics approaches with databases used in Fig 6.

Proofreading of English is needed

23: cell migration and invasion -> cell migration; cell invasion

29: <32% -> less than one third

30: suggestions of mechanisms have been made also by others

321: add here the company for Fibronectin antibody (it is now xxx)

Reviewer 3 Report

In this manuscript by Lee at. al., the authors have investigated the mechanism of ELK3-regulation of migration and invasion in gastric cancer cells. Utilizing knockdown and overexpression models, authors show that the ELK3 is required for migration phenotype. Further using transcriptional analysis, several genes were found to be regulated by ELK3 that may be implicated in cell migration and invasion. These genes were further validated in the gastric cancer patient cohorts and LOXL2 was significantly correlated with ELK3 across datasets. The main limitation of the study is lack of data showing the link between transcriptionally regulated genes and migration phenotype in cancer cells that should at least be highlighted in the discussion. While the ELK3 is already shown to be involved in gastric migration, the transcriptional regulation and identified factors is novel. I believe the manuscript is suitable for publication in IJMS, however there are few concerns that authors need to address before publication:

  1. In Fig1B and 1C, there is discrepancy in the transcript and protein levels of ELK3 in the SNU216 and SNU668 cells. Can the authors comment on this?
  2. In line 123, the authors have identified 39 ELK3 regulated genes. Further, they have finally selected 7 genes. The authors should explain how they have identified and shortlisted these 7 genes. Also, in line 130, only 6 genes are listed instead of 7.
  3. In Fig5A-5C, the authors should also show the statistical comparison between shCon and shELK3+ELK3 groups.
  4. In line 95, regulates should be corrected to regulated.

Round 2

Reviewer 1 Report

I appreciate the authors collegiality and efforts to answer all my questions and criticisms and to follow my suggestions. I believe that the answers provided along with the modifications introduced in the text to clarify methodological details have clearly contributed to improve the manuscript, bringing it to a higher level.       

Author Response

We really appreciate your kind comment. 

Thank you.

Reviewer 2 Report

The authors have addressed some of the comments adequately, but I am still missing the causal links that ELK3 would regulate migration and adhesion by regulating ECM remodeling-related genes – the major claim of the paper. It is suggested that the regulation of ECM remodeling-related genes by ELK3 is BMP1-mediated. The authors should show what the role of BMP1 is in ELK3-regulated effects (e.g. siBMP in Fig 2 setting analyzing the effect on ELK3-regulated migration and invasion).

Figure 6, apparently the data is mRNA expressions (not protein) – please clarify this in text.

Figuree 6 D, please clarify in methods, how the treshold high and low expression were determined.

Author Response

Major concerns:

  • The authors have addressed some of the comments adequately, but I am still missing the causal links that ELK3 would regulate migration and adhesion by regulating ECM remodeling-related genes – the major claim of the paper. It is suggested that the regulation of ECM remodeling-related genes by ELK3 is BMP1-mediated. The authors should show what the role of BMP1 is in ELK3-regulated effects (e.g. siBMP in Fig 2 setting analyzing the effect on ELK3-regulated migration and invasion).

We really appreciate your comments. We further analyzed the molecular link between ELK3 and BMP1-ECM remodeling-related genes to control cell migration and invasion and added new data as n figure 6. Please find the yellow highlight in the revised manuscript. Please find the yellow highlight in the revised manuscript.

  • Figure 6, apparently the data is mRNA expressions (not protein) – please clarify this in text.

We corrected the gene names in the italic font in the Figure 7 (old version of Fig 6). Please find the Figure 7 in the 2nd revised manuscript.

  • Figure 6 D, please clarify in methods, how the threshold high and low expression were determined.

We analyzed the survival curve of the patients with gastric cancer using website (https://kmplot.com/analysis/). They provided us the information about cut-off point; all possible cut-off values between the lower quartile and upper quartile are calculated, and the best performing threshold is used as a cutoff *.

*Lanczky, A. & Gyorffy, B. Web-Based Survival Analysis Tool Tailored for Medical Research (KMplot): Development and Implementation. J Med Internet Res 23, e27633 (2021).
